# Diverse Marinimicrobia bacteria may mediate coupled biogeochemical cycles along eco-thermodynamic gradients

Alyse K. Hawley [1], Masaru K. Nobu[2,3], Jody J. Wright[1], W. Evan Durno[4], Connor Morgan-Lang[4], Brent Sage[4], Patrick Schwientek[5], Brandon K. Swan[6,11], Christian Rinke [7], Monica Torres-Beltrán[1], Keith Mewis[8], Wen-Tso Liu[2], Ramunas Stepanauskas [6], Tanja Woyke [5] & Steven J. Hallam [1,4,9,10]

Microbial communities drive biogeochemical cycles through networks of metabolite exchange that are structured along energetic gradients. As energy yields become limiting, these networks favor co-metabolic interactions to maximize energy disequilibria. Here we apply single-cell genomics, metagenomics, and metatranscriptomics to study bacterial populations of the abundant "microbial dark matter" phylum Marinimicrobia along defined energy gradients. We show that evolutionary diversification of major Marinimicrobia clades appears to be closely related to energy yields, with increased co-metabolic interactions in more deeply branching clades. Several of these clades appear to participate in the biogeochemical cycling of sulfur and nitrogen, filling previously unassigned niches in the ocean. Notably, two Marinimicrobia clades, occupying different energetic niches, express nitrous oxide reductase, potentially acting as a global sink for the greenhouse gas nitrous oxide.

[1] Department of Microbiology and Immunology, University of British Columbia, Vancouver, BC V6T 1Z3, Canada. [2] Department of Civil and Environmental Engineering, University of Illinois at Urbana-Champaign, 205 North Mathews Avenue, Urbana, IL 61801, USA. [3] Bioproduction Research Institute, National Institute of Advanced Industrial Science and Technology (AIST), Central 6, Higashi, Tsukuba, Ibaraki 305-8566, Japan. [4] Graduate Program in Bioinformatics, University of British Columbia, Vancouver, BC, Canada. [5] Department of Energy Joint Genome Institute, Walnut Creek, CA 94598, USA. [6] Bigelow Laboratory for Ocean Sciences, East Boothbay, ME 04544, USA. [7] Australian Centre for Ecogenomics, University of Queensland, St Lucia, Brisbane, 4072 QLD, Australia. [8] Genome Science and Technology Graduate Program, University of British Columbia, Vancouver, BC, Canada. [9] ECOSCOPE Training Program, University of British Columbia, Vancouver, BC, Canada. [10] Peter Wall Institute for Advanced Studies, University of British Columbia, Vancouver, BC V6T 1Z2, Canada. [11]Present address: National Biodefense Analysis and Countermeasures Center, Frederick, MD 21702, USA. Alyse K. Hawley and Masaru K. Nobu contributed equally to this work. Correspondence and requests for materials should be addressed to S.J.H. (email: shallam@mail.ubc.ca)

The laws of thermodynamics apply to all aspects of Life, governing energy flow in both biotic and abiotic regimes. Nicholas Georgescu–Roegen was the first to directly apply the laws of thermodynamics to economic theory, bringing to the forefront the reality of limited natural resources on sustainable growth[1]. Robert Ayers used the term "eco-thermodynamics" to describe the application of thermodynamics and energy flow to economic models with the controversial conclusion that future economic growth necessitates the recycling of goods[2]. Within microbial ecology there is an emerging consensus that these same organizing principles structure microbial community interactions and growth with feedback on global nutrient and energy cycling[3–6]. Indeed, recycling in the common sense may be analogous to metabolite exchange or use of public goods[7], as the goods from one production stream become available for growth of another. Microbial communities living near-thermodynamic limits where high potential electron acceptors are scarce tend to utilize differential modes of metabolic coupling including obligate syntrophic interactions, maximizing any chemical disequilibria to yield energy for growth[8,9]. Thus, the term eco-thermodynamics takes on new meaning in the context of microbial ecology where thermodynamic constraints directly shape the structure and activity of microbial interaction networks.

Eco-thermodynamic gradients are formed by the distribution of available electron donors and acceptors within the physical environment, creating metabolic niches that are occupied by diverse microbial partners playing recurring functional roles[10,11]. Marine oxygen minimum zones (OMZs) provide a vivid example of eco-thermodynamic gradients shaping differential modes of metabolic coupling at the intersection of carbon, nitrogen, and sulfur cycling in the ocean[12,13]. For example, OMZ microbial communities manifest a modular denitrification pathway that links reduced sulfur compounds to nitrogen loss and nitrous oxide ($N_2O$) production[12,14–16]. While many of the most abundant interaction partners are known, recent modeling efforts point to a novel metabolic niche for the terminal step in the denitrification pathway (nitrous oxide reduction to dinitrogen gas) occupied by unidentified community members[5]. By defining the interaction networks coupling microbial processes along eco-thermodynamic gradients it becomes possible to more accurately model nutrient and energy flow at ecosystem scales.

Recent advances in sequencing technologies have opened a genomic window on uncultivated microbial diversity, illuminating the metabolic potential of numerous candidate divisions also known as microbial dark matter (MDM)[17–20]. Many MDM organisms occupy low-energy environments, where they appear to form obligate metabolic dependencies that could help explain resistance to traditional isolation methods. Marinimicrobia (formerly known as Marine Group A and SAR406) is an MDM phylum with no cultured representatives that is prevalent in the ocean. Marine Marinimicrobia have been previously implicated in sulfur cycling via a polysulfide reductase gene cluster[21,22]. In studies of a methanogenic bioreactor, Marinimicrobia have also been identified to rely on syntrophic interactions with metabolic partners to accomplish degradation of amino acids[23]. The global distribution of Marinimicrobia clades implicates a much wider diversity of both metabolic functions and partners than currently described. Here we use shotgun metagenomics, metatranscriptomics and single-cell genomics to investigate energy metabolism within the Marinimicrobia to reveal novel modes of metabolic coupling with important implications for nutrient and energy cycling in the ocean.

## Results

### Marinimicrobia single-cell amplified genomes and phylogeny.
A total of 25 Marinimicrobia single-cell amplified genomes (SAGs) from sources along eco-thermodynamic gradients were identified globally by flow sorting, whole-genome amplification and sequencing (Supplementary Data 1). SAG de novo assemblies ranged in size from 0.39 to 2.01 million bases (Mb) with estimated genome completeness ranging from <10% to >90% (average 45%) (Supplementary Table 1). Most Marinimicrobia SAGs manifested streamlined genomes, with high coding base percentage (89.99–97.13%) and low cluster of orthologous group (COG) redundancy (1.08–1.16) (Supplementary Fig. 1). PhyloPhlAn analysis of conserved marker genes placed Marinimicrobia SAGs within the bacterial domain branching deeply from the closest cultured thermophilic representative *Caldithrix abyssi* (Supplementary Fig. 2). To determine phylogenetic diversity within the Marinimicrobia, we constructed a comprehensive SSU rRNA gene tree resolving 17 clades (Fig. 1). SAG sequences were affiliated with 10 clades spanning the entire breadth of the Marinimicrobia tree (Figs. 1 and 2a, b) providing a broad phylogenetic range with which to assess distribution patterns and energy metabolism within the phylum.

**Biogeography of Marinimicrobia clades.** Using this phylogenetic information, we determined the global biogeographic distribution of Marinimicrobia and specific SAG-affiliated clades along eco-thermodynamic gradients spanning oxic (>90 μmol $O_2$), dysoxic (20–90 μmol $O_2$), suboxic (1–20 μmol $O_2$), anoxic (<1 μmol $O_2$), sulfidic and methanogenic conditions. Estimates of Marinimicrobia total abundance and clade distribution were carried out by a robust survey of 594 globally sourced metagenomes (549 assembled Illumina data sets and 45 unassembled 454 data sets) across terrestrial and marine ecosystems, including Northeastern Subarctic Pacific (NESAP, n = 43), Saanich Inlet (SI, n = 90), Eastern Tropical South Pacific (ETSP, n = 6), Peruvian (n = 17), and Guaymas Basin (n = 2) OMZs; TARA Oceans (n = 243) and several other marine (n = 141) and terrestrial sites (n = 52), (Supplementary Data 2) totaling 127 Gigabases (Gb) of sequence information. To estimate total abundance, we used a sequence similarity recruitment with a cutoff of >70% nucleotide identity over >70% of the metagenomic contig. Globally we recovered 1.3 Gb of Marinimicrobia-affiliated sequence or 1.3 million genome equivalents (assuming 1 Mb average genome) representing ~1% of surveyed data. The recovery of Marinimicrobia-affiliated sequences was highest in coastal OMZs, increasing in relation to decreasing $O_2$ concentration (Supplementary Fig. 3A). Recovery was more variable in other marine locations and minimal in terrestrial locations. To more fully resolve this sequence information at the level of specific Marinimicrobia clades, we conducted a more stringent recruitment of >95% nucleotide identity across >200 bp intervals (Supplementary Data 4). On a global scale three clades constituted 75% of observed Marinimicrobia with the remaining seven clades making up the difference (Supplementary Fig. 3B). Consistent with previous results, predominantly marine sites were recruited with two hits from terrestrial locations. Sakinaw Lake, a meromictic lake with high methane concentrations[19], was the only geographic location with recruitment to the HMTAb91 clade. Within marine systems, SAGs recruited sequences from cognate environments and conditions consistent with observed tree branching patterns (Fig. 2a–c; Supplementary Datas 3 and 4). Overall, trends indicated that specific clades inhabit particular energetic niches with potential for metabolic coupling within a given niche.

**Population genome bin construction.** To determine the energy metabolism of Marinimicrobia clades and overcome low genome completion of some SAGs, we leveraged extensive metagenomic

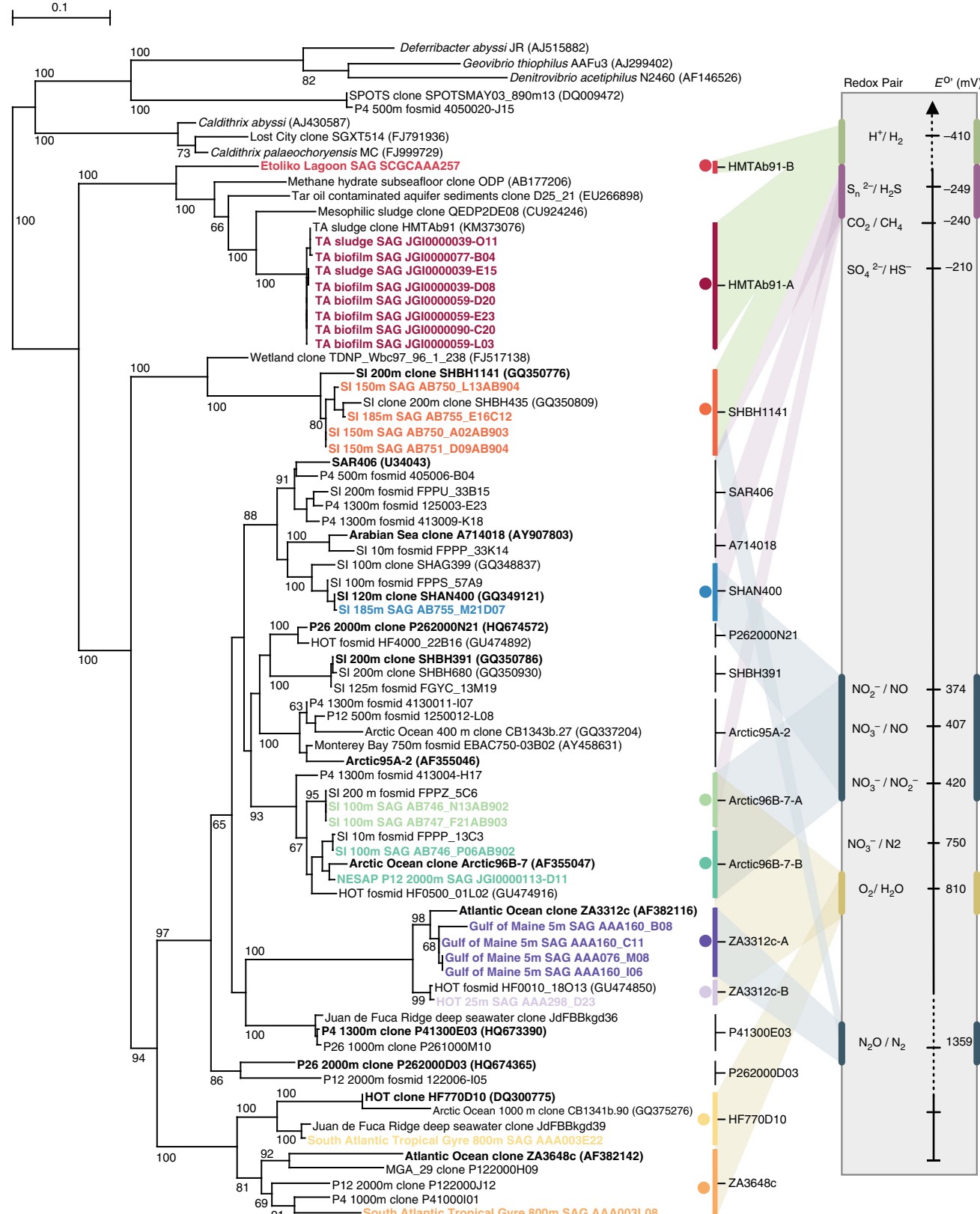

**Fig. 1** Maximum-likelihood small subunit rRNA gene tree and proposed energy metabolism for Marinimicrobia clades. Maximum likelihood phylogenetic tree of small subunit ribosomal rRNA (SSU rRNA) genes from all available studies. SSU rRNA genes from SAGs used in this study are in bold and colored to indicate there membership to population genome bins. Redox pairs are colored consistent with Fig. 1. Energy metabolism redox pairs for each clade explored in this publication are mapped to the electron tower on the right of the tree. The bar represents 1% estimated sequence divergence. Bootstrap values below 50% are not shown

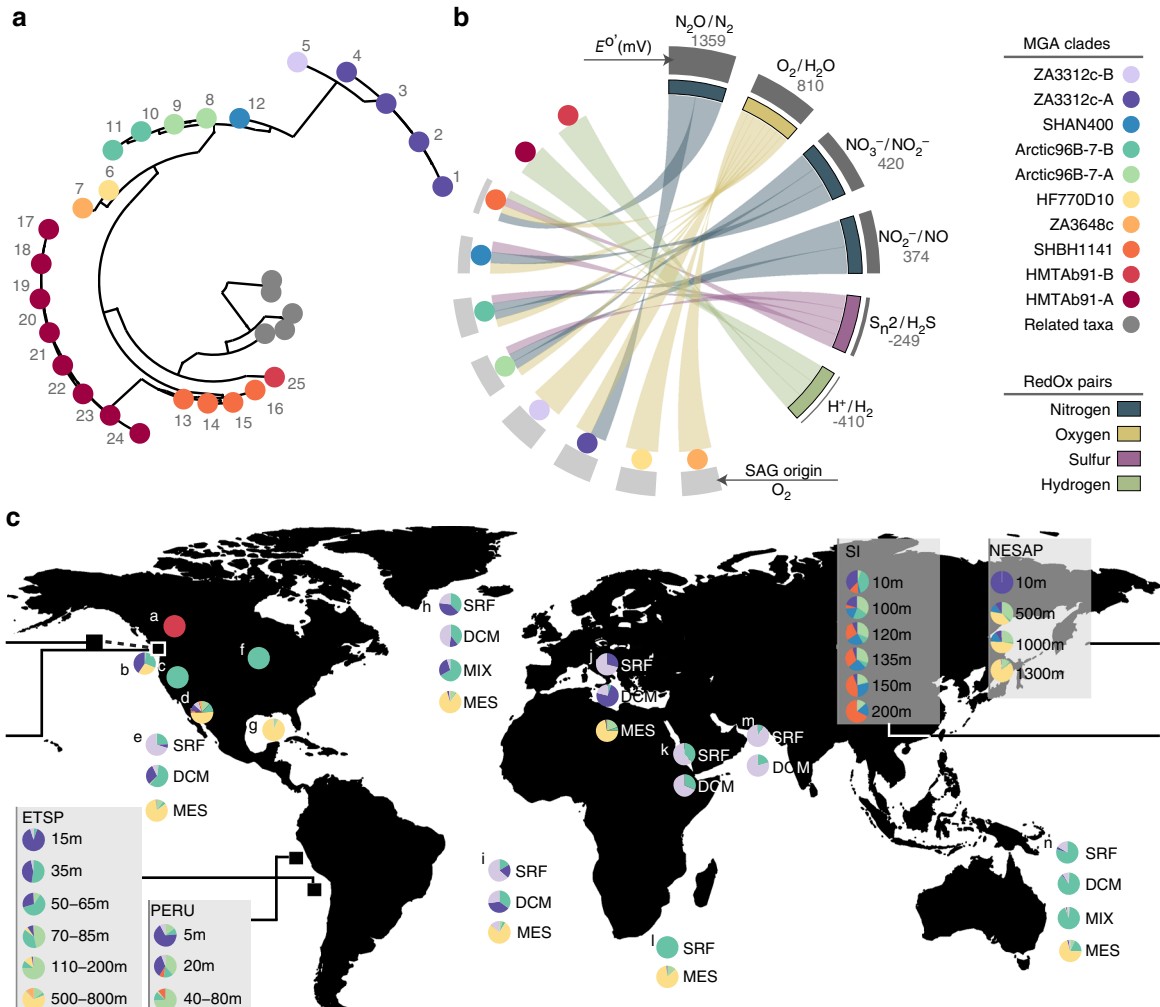

**Fig. 2** Phylogeny and biogeography of Marinimicrobia single-cell-amplified genomes and clades. **a** Unrooted phylogenetic tree based on SSU rRNA genes showing the phylogenetic affiliation of Marinimicrobia SAGs. Each dot represents a SAG in Supplementary Table 1 with the corresponding number. The tree was inferred using maximum likelihood implemented in PhyML. **b** Circular plot indicating the terminal electron acceptors used and their respective $E^{o'}$(mV) value (right) by the different Marinimicrobia clades (left). **c** Global distribution of Marinimicrobia SAG-affiliated clades, as determined by metagenomic fragment recruitment using FAST (23) with 594 global metagenomes with a threshold of ≥95% nucleotide sequence identity and alignments ≥200 bp. Recruited contig lengths were normalized by the length of each SAG assembly in mega base pairs (Mbp) and to the size of the metagenome of origin in Mbp

and metatranscriptomic resources from NESAP and Saanich Inlet time series[24, 25] to construct population genome bins, improving estimated genome completion to an average of 87% (Supplementary Data 5). Metagenomic contigs >5000 bp and with >95% identity to SAGs were identified followed by tetra-nucleotide frequency analysis to resolve specific clades (Fig. 3a). A total of five population genomes for Marinimicrobia clades ZA3312c-A/B, HF770D10, Arctic96B-7-A/B, SHAN400, and SHBH1141 spanning oxic, dysoxic, suboxic, anoxic, and anoxic–sulfidic conditions were resolved from Saanich Inlet and NESAP metagenomes, enabling more complete metabolic reconstruction within each clade (Fig. 3a, b). A sixth clade (HMTAb91-A), endemic to a methanogenic bioreactor branching near the base of Marinimicrobia radiation was included in downstream comparisons of metabolic potential to encompass the complete range of electron donor–acceptor pairs. Energy metabolism of Marinimicrobia population genomes was examined in relation to tree branching patterns and environmental disposition. A total of 18 metatranscriptomes from six depths and three time points (Fig. 4b) were used to explore Marinimicrobia gene expression over defined energy gradients including a deep

water renewal event resulting in the influx of oxygenated nutrient rich waters in Saanich Inlet basin waters. This enabled the resolution of metabolic niches and indicted potential modes of metabolic coupling within specific Marinimicrobia clades.

**Metabolic reconstruction and gene model validation.** Marinimicrobia clades ZA3312c–A/B and HF770D10 were most abundant under oxic water column conditions with extensive genome streamlining comparable to *Ca. Pelagibacter* (Supplementary Fig. 1A). All three clades harbored genes encoding for aerobic respiration, and heterotrophy with no indication for autotrophic $CO_2$ fixation. ZA3312c clades also encoded the oxidative tricarboxylic acid (TCA) cycle (Supplementary Data 6) and proteorhodopsin, a proton-pump used to harness light energy (Fig. 3b)[26]. ZA3312c proteorhodopsin transcripts were highly expressed in oxic surface waters of Saanich Inlet, suggesting that ZA3312c are capable of supplementing organotrophy with phototrophy in surface waters, a trait well suited to open-ocean oligotrophic environments (Supplementary Fig. 6A). Interestingly, ZA3312c-A encoded nitrous oxide reductase (*nozZ*)

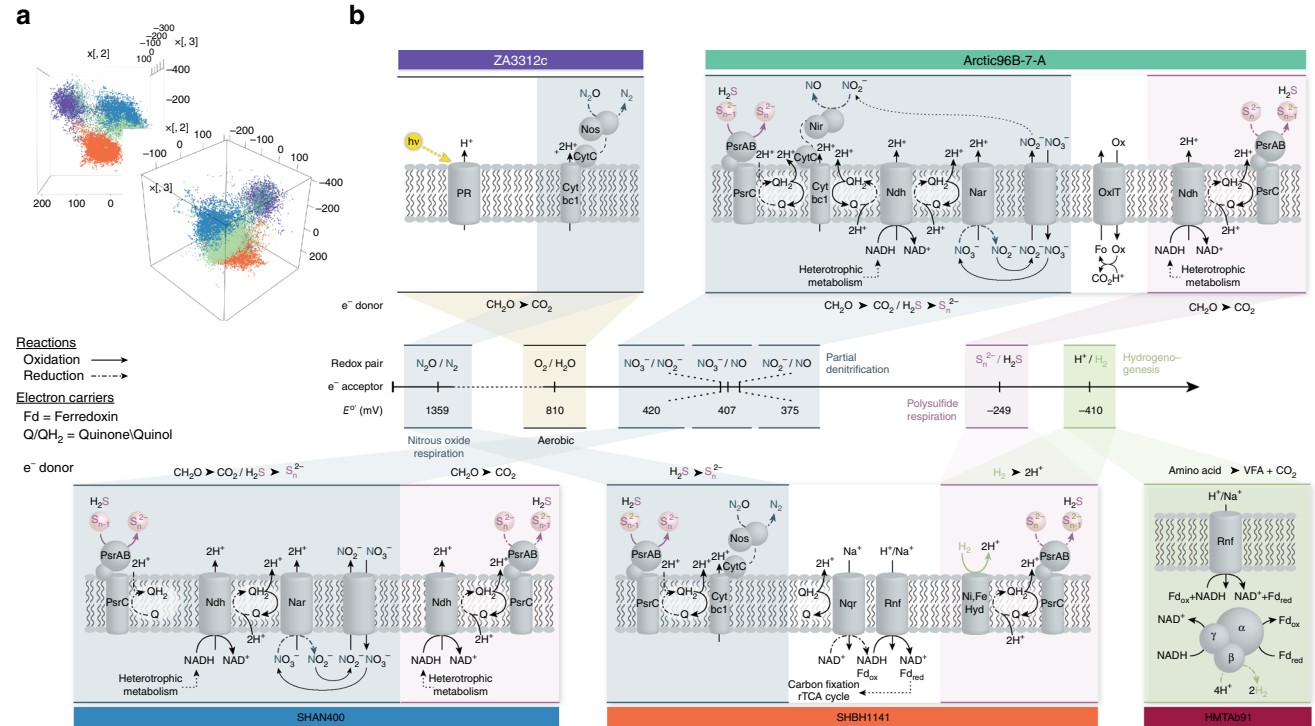

**Fig. 3** Energy metabolism of Marinimicrobia population genome bins. **a** Binning of Marinimicrobia population genomes by Kmer frequency principal component analysis, two rotations of three-dimensional plot, clouds of color coded genome bins are apparent. **b** Summary of co-metabolic and energy metabolism and conservation strategies of Marinimicrobia population genomes from along eco-thermodynamic gradients, for oxygen (beige), nitrogen (blue), sulfur (pink), and hydrogen (green). Enzymes include: proteorhodopsin (PR), sulfur: polysulfide reductase (PsrAB, PsrC); nitrogen: nitrite reductase (Nir), nitrate reductase Nar, nitrate/nitrite antiporter (NirK), nitrous oxide reductase (Nos); hydrogen metabolism: Ni,Fe hydrogenase (Ni,Fe Hyd), hydrogenase complex (HydBD); respiratory elements: cytochrome bc1 complex (Cytbc1), NADH dehydrogenase (Ndh), energy-conserving putative electron transfer mechanisms putative ion-translocating ferredoxin:NADH oxidoreductase (IfoAB); oxalate transporter (OxlT); *Rhodobacter* nitrogen fixation complex (Rnf). Oxidation and reduction indicated by solid or dotted arrows, respectively

and associated maturation factors (*nosL*, *nosD*, and *nosY*) that drive the conversion of $N_2O$ to $N_2$ in the terminal step of denitrification. Transcripts for *nosZ* were expressed throughout the Saanich Inlet water column (Fig. 4a; Supplementary Fig. 7) and indicate potential coupling to ammonia oxidizing *Thaumarchaea* that produce $N_2O$ as a byproduct of ammonia oxidation[27]. ZA3312c-A *nosZ* transcripts were also detected in suboxic waters of the NESAP, Peru, and ETSP OMZs, and four TARA oceans metagenomes contained ZA3312c-A *nosZ* sequences (>80% nucleotide identity) (Fig. 4b) reinforcing a global distribution pattern with functional implications for marine nitrogen budgets and greenhouse gas cycling. Marinimicrobia clades Arctic96B-7-A and B were widespread in dysoxic ocean waters. Arctic96B-7 clades harbored genes encoding for aerobic respiration, organotrophy and oxidative TC) cycle with no indication for proteorhodopsin or autotrophic $CO_2$ fixation (Supplementary Data 6). Arctic96B-7 clades may supplement energy generation in a similar manner to proteorhodopsin through catabolism of the common ocean compound oxalate[28], coupling a unique oxalate: formate antiporter and oxalate decarboxylase[29]. The Arctic96B-7-A clade also encoded nitrate reductase (*narG*), and polysulfide (polyS) reductase (*psrABC*) (Figs. 2 and 3b; Supplementary Figs. 6A and 8) that were expressed throughout the Saanich Inlet water column. Peak expression corresponded to depths with low $NO_3^-$ and no detectable $H_2S$ (Fig. 4a; Supplementary Fig. 6A). Interestingly, the PsrABC enzyme complex can use $H_2S$ as an auxiliary electron donor through PsrABC-mediated $H_2S$ oxidation to polyS and stored polyS can serve as an alternative electron sink, regenerating $H_2S$. The combination of *narG* and *psrABC* provides Arctic96B-7 clades with versatile energy

metabolism with potential coupling to both sulfur oxidizing bacteria (ARCTIC96-BD19, SUP05) by regenerating $H_2S$ under non-sulfidic conditions, and anaerobic ammonium (*Planctomycetes*) and nitrite (*Nitrospina*) oxidizing bacteria through the production of $NO_2^-$ in dysoxic, suboxic, and anoxic waters (Fig. 5a). Thus, Arctic96B-7 clades may form supportive metabolic partnerships with major primary producers in OMZs critical to the biogeochemical cycling of carbon, nitrogen, and sulfur[12].

Marinimicrobia clade SHAN400 appears to be endemic to Saanich Inlet where it is most abundant below the oxycline (Supplementary Fig. 4). SHAN400 harbored genes encoding for aerobic and anaerobic respiration, heterotrophy and oxidative TCA cycle. SHAN400 also encoded ferredoxin, pyruvate metabolism, and NADH dehydrogenase (Fig. 3b; Supplementary Figs. 8 and 9), potentially providing additional electron shuttles for energy metabolism under anoxic conditions. Similar to Arctic96B-7, SHAN400 encoded *narG* and *psrABC*, potentially linking its energy metabolism to both sulfur-oxidizing bacteria (SUP05) and anaerobic ammonium- (*Planctomycetes*) and nitrite- (*Nitrospina*) oxidizing bacteria in anoxic waters (Figs. 3 and 4; Supplementary Fig. 6A, B). In contrast to Arctic96B-7, SHAN400 transcripts for heme/copper-type cytochrome and NADH dehydrogenase were most highly expressed in anoxic waters (Supplementary Fig. 9A). This is consistent with redox-driven niche partitioning between Arctic96B-7 and SHAN400 clades in the Saanich Inlet water column.

Marinimicrobia clade SHBH1141 was prevalent in anoxic and anoxic–sulfidic OMZ waters (Supplementary Fig. 4). SHBH1141 harbored genes encoding for aerobic and anaerobic respiration, autotrophic $CO_2$ fixation via the reductive TCA cycle (citrate

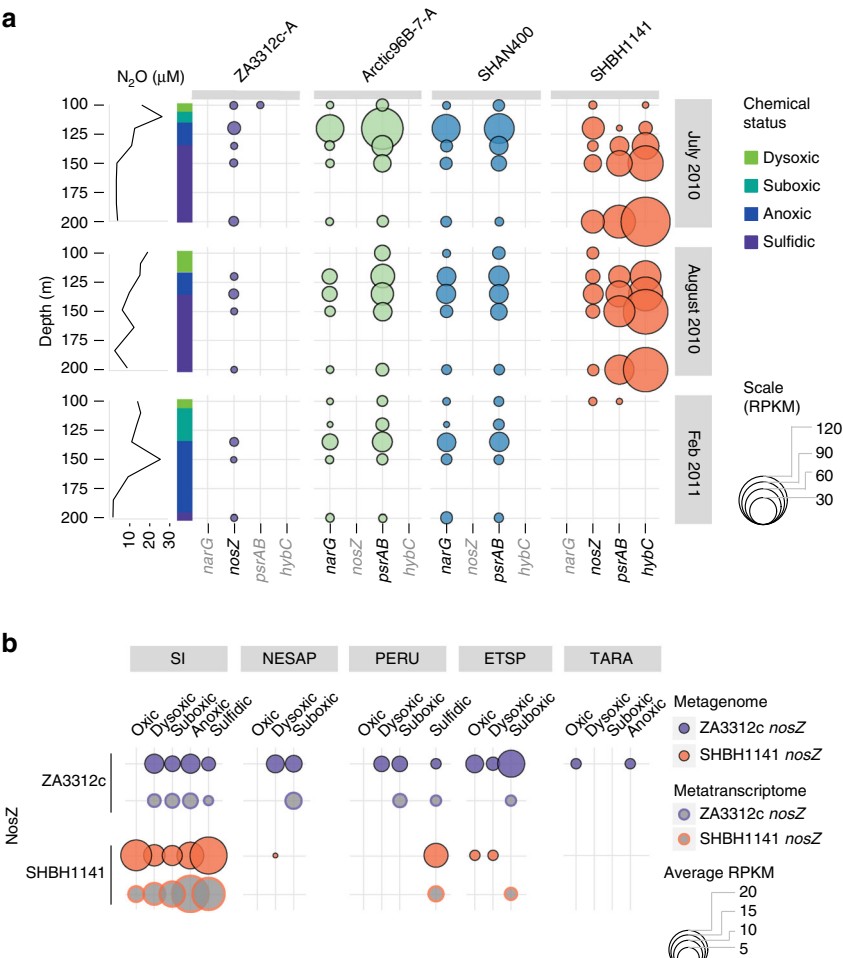

**Fig. 4** Expression of selected Marinimicrobia energy metabolism genes. **a** Expression of selected genes involved in Marinimicrobia energy metabolism in Saanich Inlet station SI03 at three time points and five depths between 100 and 200 m. Size of circle represents reads per kilobase per million mapped (RPKM)[52] for metatranscriptomic reads mapped to the selected genes for the indicated population genomes. Water column redox state for each time point encoded on left axis and nitrous oxide concentration profile for each time point on left. Enzymes: nitrate reductase (narG), Nitrous oxide reductase (nosZ), polysulfide reductase subunits A and B (psrAB) and Ni-Fe hydrogenase subunits A and B (hybC). **b** Detected genes and transcripts for Marinimicrobia ZA3312c and SHBH1141 nosZ along eco-thermodynamic gradients from oxic (>90 μmol $O_2$), dysoxic (20–90 μmol $O_2$), suboxic (1–20 μmol $O_2$), anoxic (<2 μmol $O_2$), and sulfidic conditions in Saanich Inlet (SI) time series, Northeastern Subarctic Pacific (NESAP), Peru, Eastern Tropical South Pacific (ETSP), and TARA Oceans (no transcriptomes available) data sets. For SI and ETSP dot size represents average reads per killobase per million mapped (RPKM) summed for a given nosZ type for each metagenome or metatranscriptome and averaged by the total number of metagenomes or metatranscriptomes for a given water column classification. For ETSP, Peru, and TARA bubble size is the number of reads (ETSP and Peru) or contigs (TARA) with nosZ averaged per number of metagenome or metatranscriptomes for a given water column classification

lyase and ferredoxin-dependent 2-ketoacid oxidoreductases), and the *Rhodobacter* nitrogen fixation (Rnf) complex to produce reduced ferredoxin to drive endergonic reductive carboxylation steps, indicating a capacity to perform anaerobic autotrophy (Supplementary Figs. 8 and 9). In addition, SHBH1141 encoded *psrABC*, class I [Ni,Fe] hydrogenases (*hybOABCD*) and *nosZ* with associated maturation factors *nosL* and *nosD* (Fig. 3b; Supplementary Figs. 6A and 8). Gene expression for *psrABC*, *hybOABCD*, and *nosZ* was elevated under anoxic to sulfidic conditions (120 m in July 2010, and 150 m in July and August 2010; Fig. 4). SHBH1141 class I [Ni,Fe] hydrogenase is proposed to operate bidirectionally based on observations in *Escherichia coli* and *Salmonella enterica*, with proposed hydrogen production under more oxidizing conditions[30]. SHBH1141 *nosZ* was recovered on a global scale and expressed under both sulfidic conditions in Peru and suboxic conditions in the ETSP as well as Saanich Inlet (Fig. 4b), positing a central role for SHBH1141 in OMZ $N_2O$ reduction. The expression of these genes in

anoxic–sulfidic waters points to a new mode of dynamic metabolic mutualism in which SHBH1141 may rely on SUP05 $N_2O$ generation in anoxic and sulfidic waters[12,31] to store polyS and re-evolve $H_2S$ from polyS to stimulate SUP05 $N_2O$ production (Fig. 5b). This would in turn support autotrophic carbon fixation in both partners and sustains N and S biogeochemical cycling under dynamic or unfavorable conditions (e.g., limited $H_2S$ bioavailability; Supplementary Fig. 5). Such mutualism would be highly dependent on either (a) migration along the eco-thermodynamic gradient or (b) seasonal/temporal changes such as renewal or upwelling events.

Marinimicrobia clades HMTAb91-A/B are prevalent in methanogenic locations at the base of the electron tower. Apparently, HMTAb91-A/B did not harbor genes for aerobic respiration and had an incomplete TCA cycle. HMTAb91-A encoded the Embden–Meyerhof–Parnas pathway (Supplementary Data 6) and both HMTAb91-A/B encoded energy-conserving $H^+$ respiration through electron-confurcating hydrogenases, the

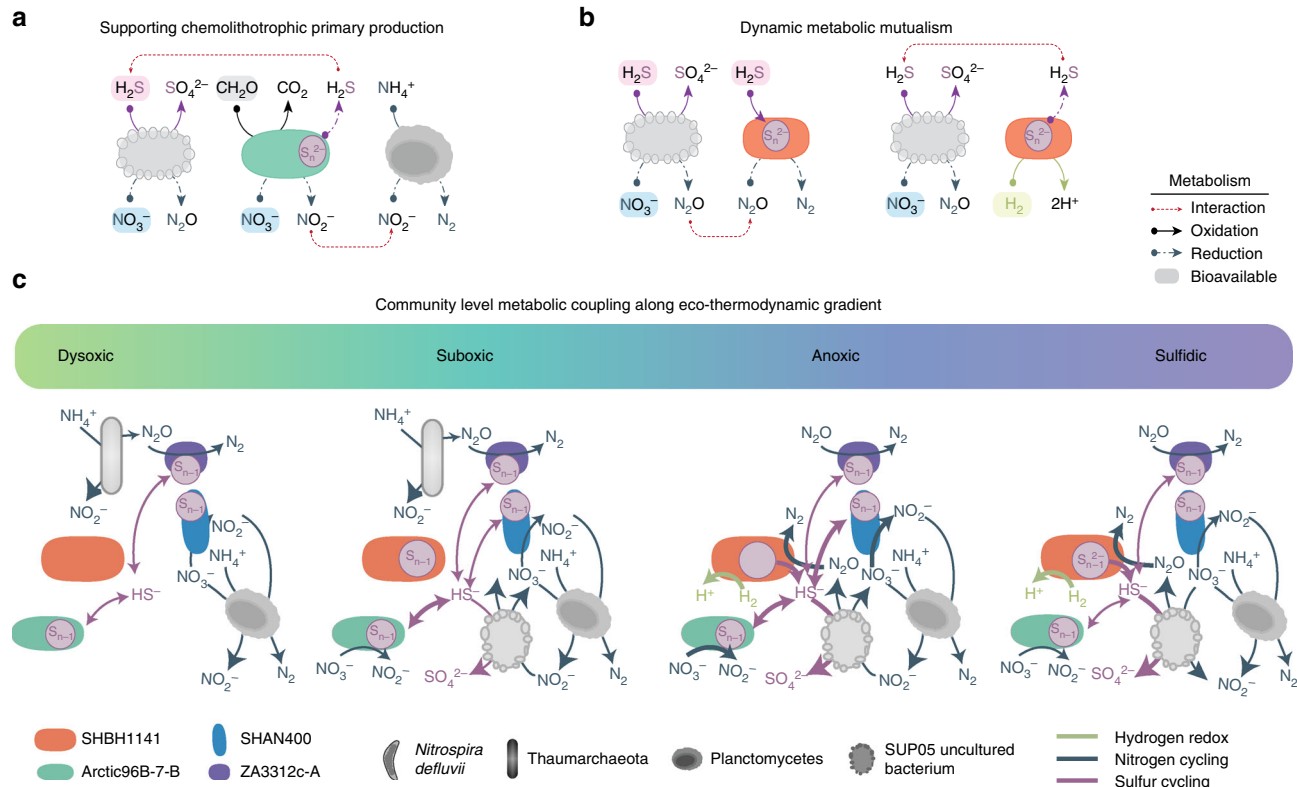

**Fig. 5** Proposed co-metabolic model along eco-thermodynamic gradient in Saanich Inlet. **a** Proposed metabolic coupling between ARCTIC96B-1, SUP05, and Planctomycetes. **b** Proposed dynamic metabolic mutualism between SUP05 and SHBH1141. **c** Conceptual model for Marinimicrobia co-metabolic activity with other major microbial groups in Saanich Inlet along eco-thermodynamic gradients. Interactions based on expression data for sulfur (pink), nitrogen (blue), and hydrogen (green) for dominant Marinimicrobia clades in Saanich Inlet as well as putative metabolic partners *Nitrosopumulaceae* sp., *Planctomycetes*, and SUP05

energy-conserving (Rnf complex) and putative syntrophic amino-acid metabolism through the ion-translocating ferredoxin:NADH oxidoreductase (*ifoAB*) (Fig. 3b)[23]. Within the methanogenic reactor where it was initially described, HMTAb91-A is postulated to accomplish thermodynamically unfavorable amino-acid degradation supporting methanogenesis[23]. HMTAb91-A/B clades appear restricted to methanogenic eco-systems as no metagenomic or metatranscriptomic sequences were recruited from non-methanogenic locations.

## Discussion

Co-metabolic functions encoded and expressed within globally distributed Marinimicrobia clades would fill several hitherto unassigned niches in the nitrogen and sulfur cycles and support recent modeling efforts integrating biogeochemical and multi-omic sequence information in the Saanich Inlet water column[24,32,33] (Fig. 5). The $N_2O$ reductase expressed on a global basis by ZA3312c-A and SHBH1141 clades has the potential to act as a biological filter for $N_2O$ produced by the ubiquitous marine processes of ammonia oxidation (e.g., *Thaumarchaeota*)[27] and partial denitrification (e.g., SUP05)[12,31]. In contrast, nitrate reduction to $NO_2^-$ by other Marinimicrobia clades (i.e., Arctic96B-7-A and SHAN400) has potential to provide $NO_2^-$ to anaerobic ammonium-oxidizing (*Planctomycetes*) and nitrite-oxidizing (*Nitrospina*) bacteria in dysoxic, suboxic, and anoxic waters. The polysulfide reductase expressed by multiple Marinimicrobia clades (e.g., Arctic96B-7, SHAN400, and SHBH1141) has potential to provide an energy storage mechanism via accumulation of polyS that can be reduced or oxidized under changing water column redox conditions and support both

cooperative and dynamic interactions including cryptic sulfur cycling and dark carbon fixation[34].

The application of eco-thermodynamics principles to microbial ecology provides perspective on how thermodynamic constraints serve to shape microbial community structure and the nature of co-metabolic interactions along energy gradients. Indeed, phylogenetic branching patterns often coincided with energy yields of redox pairs for identified clade energy metabolism, with deeper branching clades near the base of the electron tower where lower energy yields would increase potential for metabolic coupling. Additionally, many Marinimicrobia clades encoded enzyme systems tied to both nitrogen- and sulfur-cycling, suggesting extensive specialization for metabolic cooperation bridging within and between biogeochemical cycles. Such dependencies likely confound isolation efforts within the phylum and point to an ancestral state primed for co-existence. The extent to which this reflects the diversification of other phyla, particularly MDM across the Tree of Life is an interesting area of research with implications for understanding and directing the evolution of metabolic networks driving Earth's biogeochemical cycles.

## Methods

**SAG collection, sequencing, assembly, and decontamination**. SAGs from Gulf of Maine, HOT station ALOHA, South Atlantic Gyre, the Terephthalate degrading bioreactor and Etoliko Lagoon Sediment were included in Rinke et al.[17], and collection, assembly and decontamination follows accordingly. See Supplementary Data 1 for details on SAG genomics. SAGs from Northeast subarctic Pacific (NESAP) and Saanich Inlet followed the following protocol. Replicate 1-ml aliquots of sea water collected for single-cell analyses were cryopreserved with 6% glycine betaine (Sigma-Aldrich), frozen on dry ice and stored at −80 °C. Single-cell sorting, whole-genome amplification, real-time PCR screens, and PCR product sequence analyses were performed at the Bigelow Laboratory for Ocean Sciences Single Cell

Genomics Center (www.bigelow.org/scgc), as described by Stepanauskas and Sieracki[35]. SAGs from the NESAP were generated at the DOE Joint Genome Institute (JGI) using the Illumina platform as described in Rinke et al.[17]. SAGs from Saanich Inlet were sequenced at the Genome Sciences Centre, Vancouver BC, Canada, as described in Roux et al.[36]. All SAGs were assembled at JGI as described in Rinke et al.[17,36].

The following steps were performed for SAG assembly: (1) filtered Illumina reads were assembled using Velvet version 1.1.04[37] using the VelvetOptimiser script (version 2.1.7) with parameters: (--v --s 51 --e 71 --i 4 --t 1 --o "-ins_length 250 -min_contig_lgth 500") 2) wgsim (-e 0 −1 100 −2 100 -r 0 -R 0 -X 0) 3) Allpaths-LG (prepareAllpathsParams: PHRED_64 = 1 PLOIDY = 1 FRAG_COVERAGE = 125 JUMP_COVERAGE = 25 LONG_JUMP_COV = 50, runAllpathsParams: THREADS = 8 RUN = std_pairs TARGETS = standard VAPI_WARN_ONLY = True OVERWRITE = True). SAG prediction analysis and functional annotation was performed within the Integrated Microbial Genomes (IMG) platform[38] (http://img.jgi.doe.gov) developed by the Joint Genome Institute, Walnut Creek, CA, USA.

**Phylogenomic analysis of SAGs.** The PhyloPhlAn pipeline was used to determine relationships among Marinimicrobia SAGs[39] (Supplementary Fig. 3) as well as the phylogenetic placement of Marinimicrobia within the bacterial domain (Supplementary Fig. 2). In both cases, fasta files for the 25 SAGs and related genomes were passed to PhyloPhlAn and resulting trees were visualized and drawn using GraPhlAn. The 25 Marinimicrobia SAGs and related genomes were inserted into the already built PhyloPhlAn microbial Tree of Life containing 3737 genomes using the "insert" functionality, and a de novo phylogenetic tree was created using the "user" functionality based solely on the 25 Marinimicrobia SAG and related genome fasta files. Default parameters were used in each case with the exception of a custom annotation file used in GraPhlAn to colour the leaves based on phylum in the microbial Tree of Life, and subgroup in the de novo phylogenetic tree.

**Metagenome fragment recruitment.** The proportion of Marinimicrobia represented in the 594 globally distributed metagenomes (Supplementary Fig. 3A) was determined by SAG nucleotide sequence alignment to individual metagenomes using FAST[40]. Parameters of 70% nucleotide identity cutoff over 70% of the contig length (or 454-read, where applicable) were employed to encompass the Marinimicrobia phylum[41]. The small subunit ribosomal RNA (SSU rRNA) gene was removed from SAG sequences before alignment searches to prevent cross-recruitment to non-Marinimicrobia sequences. The total length of contigs passing the cutoff for a given metagenome was summed and divided by the total contig length for that metagenome to calculate percentage of Marinimicrobia. Where data on $O_2$ concentration was available, for Saanich Inlet, NESAP, ETSP[15], and Peruvian upwelling[42], $O_2$ status of the sample was used as indicated. Data on $O_2$ concentration were unavailable for Marine-Misc. and terrestrial samples.

Biogeography of Marinimicrobia SAG-affiliated clades was similarly determined using alignment parameters of 95% identity cutoff and >200 base pairs (bp) alignment length to ensure only contigs with high sequence similarity while maintaining clade resolution. Metagenomic contigs mapping to more than one Marinimicrobia clade were assigned to the clade with greatest percent identity and in the event of a tie were assigned to the clade with the greatest alignment length. Overall abundance was calculated for each metagenome by summing the total lengths of all contigs with hits to a given Marinimicrobia clade divided by the total size of the SAG and the total size of the assembled metagenome in base pairs. Results by metagenome (Supplementary Datas 2 and 3) and clade were then summed in Fig. 1c and itemized in Supplementary Data 4. Global relative abundance of Marinimicrobia clades shown in Supplementary Fig. 3B was calculated similarly by summing the total lengths of all contigs with hits to a given Marinimicrobia clade divided by the total size of the SAG and the total size of the assembled metagenome in base pairs and then summing for all hits to a given clade.

**Saanich Inlet and NESAP metagenomes and metatranscriptomes.** Saanich Inlet metagenomes and metatranscriptomes were collected, sequenced, and assembled as described in Hawley et al.[24] and cognate chemical and physical measurements can be found in and Torres-Beltran et al.[32]. Briefly, Saanich Inlet samples for metagenomic and metatranscriptomic sequencing were collected by Niskin or Go-Flow on line with CTD. Samples for metatranscriptomics, 2 l, were filtered by peristaltic pump with in-line 2.7 μM prefilter onto a sterivex filter with 1.8 ml RNALater added and frozen on dry ice within 20 min of bottle on-deck. Metagenomic samples, 20 l, were filtered within 8 h of collection by peristaltic pump with in-line 2.7 μM prefilter onto a sterivex filter with 1.8 ml lysis buffer added and frozen at −80 °C. Metagenomic and metatranscriptomic samples were processed, sequenced, and assembled according to Hawley et al.[24] at the JGI using the Illumina HiSeq platform.

Sampling in the NESAP was conducted via multiple hydrocasts using a Conductivity, Temperature, Depth (CTD) rosette water sampler aboard the *CCGS John P. Tully* during three Line P cruises: 2009-09 [June 2009, major stations P4 (48°39.0 N, 126°4.0 W, 7 June), P12 (48°58.2 N, 130°40.0 W, 9 June), and P26 (50°N, 145°W, 14 June), 2009–10 [August 2009, major stations P4 (21 August), P12 (23 August) and P26 (27 August)], and 2010-01 [February 2010, major stations P4 (4 February) and P12 (11 February)]. At these stations, large volume (20 l) samples for DNA isolation were collected from the surface (10 m), while 120 l samples were taken from three depths spanning the OMZ core and upper and deep oxyclines (500, 1000, 1300 m at station P4; 500, 1000, 2000 m at station P12). Sequencing and assembly was carried out as described above for Saanich Inlet and accession numbers are available in Supplementary Data 2.

**Construction and validation of population genome bins.** Marinimicrobia population genome bins were constructed by identifying metagenomic contigs from Saanich Inlet, and NESAP metagenomes mapping to specific SAG(s) using a supervised binning method based in part on methodologies developed by Dodsworth et al.[43] in the construction of OP9 population genome bins. Initially, determination of membership of individual SAGs to SAG-clusters making up a given phylogenetic clade was conducted. SAG tetranucleotide frequencies were then calculated and converted to z-scores with TETRA (http://www.megx.net/tetra)[44,45]. Z-scores were reduced to three dimensions with principal component analysis (PCA) using PRIMER v6.1.13[46] and hierarchical cluster analysis of the z-score PCA with Euclidian distance (also performed in PRIMER) was carried out to generate SAG-clusters. These SAG-clusters reflected phylogenetic placement of the SAGs by SSU rRNA gene analysis. For construction of population genome bins, metagenomic contigs from NESAP and SI data sets were aligned to SAG contigs with >95% nucleotide identity using BLAST[47] and a minimum of 5 kilobase pairs alignment length, Tetranucleotide frequencies of all metagenomic contigs passing this identity and length threshold were calculated and converted to z-scores. SAG-supervised binning as described in Dodsworth et al. using linear discriminant analysis was carried out using all z-scores with the SAG-bins as training data to classify the metagenomic conigs as making up a given population-genome bin.

Individual SAGs and population genome bins were analyzed for completeness and strain heterogeneity using CheckM v1.0.5[54]. Specifically, the lineage_wf workflow was used with default parameters. The lineage_wf workflow includes determination of the probable phylogenetic lineage based on detected marker genes. The determined lineage then dictates the sets of marker genes that is most relevant for estimating a given genome's completeness and other statistics. The strain heterogeneity metric is highly informative for population genome bins as it is essentially the average amino-acid identity for pairwise comparisons of the (lineage appropriate) redundant single-copy marker genes within a population genome bin (Supplementary Data 5). For population genome bins the higher the strain heterogeneity value, the more similar the amino acid identity of the redundant maker genes indicating the sequences in the bin originate from a closely related, if not identical, phylogenetic source.

**Marinimicrobia genome streamlining.** Gene-coding bases and COG-based gene redundancy shown in Supplementary Fig. 1A, B were calculated using cluster of orthologous group (COG)-based genome redundancy as described in Rinke et al.[17]. Each gene's COG category was predicted through the JGI IMG pipeline. COG redundancy was calculated by averaging the occurrence of each COG in the genome. The percentage of gene-coding bases was calculated by dividing the number of bases contributing to protein- and RNA-coding genes by the total genome size. For SAGs, the length of the assembled genome was used rather than the estimated genome size.

**Annotation and identification of metabolic genes of interest.** Genes of interest were identified in the SAGs and in IMG/M (https://img.jgi.doe.gov/cgi-bin/m/main.cgi)[48] for the metagenomic contigs which made up the population genome bins. Contigs making up Marinimicrobia population genome bins were run through MetaPathways 2.5[49,50] to annotate open reading frames (ORFs) and reconstruct metabolic pathways. As the population genome bins were constructed from multiple metagenomes they contained redundant sequence information, BLASTp[47] (amino-acid identity cutoff >75%) was used to identify all copies of a given gene of interest in each population genome bin, which was then used in gene model validation and expression mapping.

**Gene expression mapping.** Metatranscriptomes from three time points in Saanich Inlet time series[24] were used to investigate changes in gene expression along water column redox gradients over time for selected ORFs involved in energy metabolism and electron shuttling. Quality controlled reads from metatranscriptomes were mapped to identified ORFs of interest using bwa −mem[51] and reads per kilobase per million mapped (RPKM) per ORF was calculated using RPKM calculation in MetaPathways 2.5[52]. For each population genome bin RPKM values for a given sample were summed for ORFs with the same functional annotation to yield an RPKM for a given functional gene. For other taxonomic groups in Saanich Inlet shown in Supplementary Fig. 6B, genes were identified by sequence alignment searches of Saanich Inlet metatranscriptomes (bioSample indicated above) assembled and conceptually translated using BLASTp against selected nitrogen and sulfur cycling genes from Hawley et al.[12] and RPKM values calculated as described above.

**Global distribution and expression of nosZ.** Further analysis was carried out to determine the global distribution of Marinimicrobia *nosZ* in 594 metagenomes. The *nosZ* nucleotide sequences from SHBH1141 and ZA3312c, which exhibited a 65% nucleotide identity to each other by BLAST, were clustered at 95% identity using the USEARCH cluster fast algorithm[53], resulting in three clusters, two SHBH1141 and one ZA3312c. Nucleotide sequence alignment was carried out using FAST[40], with parameters of >80% nucleotide identity and >60 bp alignment length against 594 metagenomes. For Saanich Inlet and NESAP data sets, abundance of *nosZ* in a given metagenome or metatranscriptome was determined by summing the RPKM value for ORF hits to either SHBH1141 or ZA3312c for a given metagenome or metatranscriptome. For 454 sequenced[15,42] metagenomes and metatranscriptomes (Peru[42] and ETSP[15]), the number of reads which hit to either SHBH1141 or ZA3312c were summed for a given metagenome. For the TARA Oceans data set, the number of genes identified in an assembled metagenome was summed. Metatranscriptomic data for Tara was unavailable at this time.

**Data availability.** Single-cell amplified genomes and associated assemblies generated for this study from Saanich Inlet and the northeastern subarctic Pacific Ocean are available in JGI IMG with Taxon OIDs: 2537562244, 2537562243, 2537562242, 2537562237, 2537562241, 2537562240, 2537562239, 2537562238, and 2537562245. Metagenomes from Saanich Inlet and the northeastern subarctic Pacific Ocean are available at NCBI with BioSample accession codes: SAMN0324878 to SAMN0324887, SAMN0324895 to SAMN0324900, SAMN0324919, SAMN0324920, SAMN0324964 to SAMN0324982, and SAMN0324987 to SAMN0324991. Metatranscriptomes used for expression analysis are available at NCBI with BioSample accession codes: SAMN05223291 to SAMN05223293, SAMN05224498 to SAMN05224507, SAMN05224510, SAMN05224511, SAMN05224516, SAMN05224517, and SAMN05236416.

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

## Acknowledgements

We thank the Joint Genome Institute (JGI), including Susannah Tringe, Stephanie Malfatti, and Tijana Glavina del Rio, for technical and project management assistance. We thank Captain Ken Brown and his crew for all their support aboard The *RSV Strickland*, as well as our sea-going technicians at UBC, Chris Payne and Laura Pakhomova. We thank the scientists and crew aboard CCGS *John P. Tully*, in particular Marie Robert, as well as Fisheries and Oceans Canada for logistical support. We thank the officers and crew of the RV Ka'imikai-O-Kanaloa and the HOT team for sample collection at station ALOHA, and Jane Heywood and Michael Sieracki for South Atlantic field sample collection. We thank the many technicians and undergraduate helpers in the Hallam lab for support. This work was performed under the auspices of the US Department of Energy (DOE) JGI supported by the Office of Science of US DOE Contract DE-AC02- 05CH11231, by National Science Foundation Grants OCE-1232982 (to R.S. and B.K.S.), the G. Unger Vetlesen and Ambrose Monell Foundations, the Tula Foundation-funded Centre for Microbial Diversity and Evolution, the Natural Sciences and Engineering Research Council of Canada (NSERC), the Canada Foundation for Innovation, the Canadian Institute for Advanced Research through grants awarded to S.J.H., and the US National Science Foundation grant OCE-1232982 to R.S. and B.S. J.J.W. was supported by NSERC and the Tula Foundation. M.T.-B. was supported by Consejo Nacional de Ciencia y Tecnología (CONACyT) and the Tula Foundation. A.K.H. was supported by the Tula Foundation.

## Author contributions

A.K.H. carried out biogeography analysis, expression analysis, denitrification pathway analysis, prepared most figures, and aided in energy metabolism analysis. M.K.N. aided in writing and carried out energy metabolism and genome streamlining analysis and associated figures. J.J.W. assisted in composition and carried out phylogenetic analysis and associated figure production and conception of project. W.E.D. carried out population genome bin construction and regression analyses. C.M.-L. carried out CheckM analysis and read-mapping. B.S. carried out PhyloPhlAn analysis and aided biogeography analysis. P.S. aided in S.A.G. assembly and biogeography. B.K.S. aided in SAG acquisition and biogeography analysis. C.R. aided in biogeography analysis. M.T.-B. aided in *nosZ* expression analysis. K.M. aided in production of Fig. 1b. M.-T.L. aided in energy metabolism and genome streamlining analysis. R.S. carried out S.A.G. sorting and provided feedback on analyses. T.W. aided in S.A.G. decontamination. S.J.H. designed the research, aided in data analysis, and interpretation and supervised the group. A.K.H. and S.J.H. wrote the paper with input from co-authors.
