## [Peer Review File · Nature Communications]

Reviewers' comments:

Reviewer #1 (Remarks to the Author):

Hawley et al. present an extensive omics analyses of an uncultivated bacterial lineage, Marinimicrobia. The analyses include single-cell genomics, metagenomics and metatranscriptomics and are primarily derived from oxygen minimum zones. The authors describe 10 distinct clades within the Marinimicrobia and try to identify metabolic interactions associated with each clade. Important findings in this study include the identification of a link between evolutionary diversification of Marinimicrobia and metabolic energy yields and their participation in C and N biogeochemical cycling. Finally, the authors propose that certain lineages of Marinimicrobia may express nitrous oxide reductase and thereby act as a sink for nitrous oxide in the oceans. In general, the analyses are well-done but I have strong reservations over the focus of the manuscript and especially some of the key findings.

Key concerns

My key concern is that the primary findings of a link between energy yields and the diversification of Marinimicrobia and the idea that they act as a global sink for N₂O is completely unsubstantiated. We know from a multitude of genomics studies that metabolic diversity is commonly observed in the environment, especially in microbial dark matter phyla. While I agree that this diversification could be linked to energy yields in specific niches, the authors provide no evidence of the same in Marinimicrobial lineages described in this manuscript. Part of the problem is associated with the treatment of omic approaches which is quite perplexing and could be improved. As an example, N₂O is identified as a key electron acceptor in the OMZs in two lineages, SHBH1141 and ZA3312c-A. The best genome completion estimates for these two lineages are 57% and 78%, respectively. Consequently, it is quite likely that other parts of the denitrification pathway might have been missed due to these incomplete genomes. The use of single cell genome clusters for mapping metagenomic scaffolds in lieu of directly employing an unsupervised binning approach for metagenomes like Metabat is problematic in my opinion especially if the key question centers on determining the confidence of missed genes. While a link between a certain lineage and a particular redox pair like N₂O/N₂ likely exists as determined from metabolic gene predictions, it can only be validated with further proof of (a) association with N₂O (b) disassociation with other components of the reductive nitrogen cycle like NO₃, NO₂, NO. Finally, it is an entirely different argument to suggest that evolutionary diversification is driven by specific metabolic innovations that are associated with the geochemical gradients.

With advances in microbial ecology, there is a greater appreciation for the interconnectedness of microbial communities and metabolism. Overall, I think this study and the dataset itself are teeming with potential. Yet as things stand, the biogeochemical discoveries reported need further proof to be validated and accepted.

Other concerns:

Lack of context in the introduction: Much of the introduction is spent talking about thermodynamics and the evolution of the term "eco-thermodynamics". I would have appreciated more background and context especially in line with previous findings of the abundance and sulfur cycling in Marinimicrobia.

Genome reduction and streamlining: This is an interesting idea and is recurring mention throughout the manuscript. Fig. S1 comparing genome reduction in Marinimicrobia with other organisms is smart. However estimating these numbers from incomplete SAGs raises doubts about their accuracy. This could be benchmarked by plotting results from a number of SAGs with varying genome completions and checking for the slope of the line.

Genome completion: My understanding of checkM is that it uses lineage-specific marker genes.

Considering that very few Marinimicrobia genomes existed in public databases prior to this study, an alternative set of single-copy genes to determine genome completion would be useful.

HyaAB – Fig S8.b shows the hydrogenase operon structure. This seems to be a rather unusual operon with *psrC* and *psrB* in the midst of *hyaAB*. The authors should check for the other components of the hydrogenase system. Additionally, a tree of *hyaAB* genes might also be useful.

Cometabolism – Co-metabolism is a rather unusual word to use to describe the presence of multiple metabolisms. Cometabolism is more commonly used to determine the auxiliary degradation or transformation of a metabolite by a particular enzyme.

Use transcriptomics to link organisms – The authors primarily link organisms to each other using data derived from metabolic genes. While this is a good start, the presence of transcriptomic data for other organisms (Planctomycetes, Nitrospina, SUP05) as shown in Fig. S6 might be more useful to nail down some of the identified metabolic networks.

Line 99 – 101: I think this is highly speculative. To suggest that branching patterns actually coincide with energy yields, the authors would need to show that the redox pair identified with the particular group in Fig.1 is in fact the primary mode of metabolism in the environment for that organism.

Focus on sulfur metabolism – Considering the authors have reported *psrABC* in Marinimicrobia in a previous study and no other sulfur genes were identified, the findings of sulfur cycling are not novel and could be toned down a little in the abstract and conclusions.

Title – The title as it stands is extremely vague and misleading. The authors should consider modifying it to make it more meaningful.

Line 236 – “application of economic principles to microbial ecology” I am uncertain as to what principles are referred to here.

Minor:

Line 77 – “obligate metabolic interactions” is a strong word. I would prefer “likely” or “dependence” in this scenario.

Fig.2 – Redox tower is somewhat misleading as to a reader it may seem that the tree correlates with the energy yields/redox pairs which is not the case. Please include statement saying redox pairs are colored by lineage in the caption.

Table S2 – “inventory” is spelt wrong.

Line 141 – should read fig.4

Figure legends and axes are upside-down in a number of figures (like in fig S9). Please check these again.

Reviewer #2 (Remarks to the Author):

The authors describe, in a multi-omic study, the phylogeny and metabolic framework within the Marinimicrobia phylum. The authors use these datatypes to draw links to co-metabolic interactions. While representatives within this phylum do not appear to be prevalent (<10% of all the metagenomes groups shown, Fig S3), their activities are interesting and possibly allow them to

fill fundamental niches within the marine environment.

Major:

No where do I see a clear representation of the global metagenomes recruitment distinctly showing relative abundance globally. As this is a main point to the paper it would be good to have either a table or representation for clear reference. Table S3 does not seem to have good representation and Figure S3 also collapses the datasets by type, e.g., Coastal OMZ, Open Ocean etc. And, Figure 1C, while nice to show distribution does not adequately the relative abundance of these clades.

Minor:

In the supplemental material methods section 'Metagenome fragment recruitment for biogeography' Line 65 it states that the contigs were assigned to the clade with greatest percent identity OR alignment length. What would happen if a contig had the the greatest percent identity to one clade and the alignment length to another. Where would it be assigned. This seems ambiguous or I am missing something.

Table S3: This appears to be the numbers of contigs (Illumina) or reads (454) that hit each Marinimicrobia lineage from the different metagenome groups and also appears very low. Is this the correct number ... as in 1 sequence (which could be a single gene from Illumina – it's hard to tell the contig lengths here) hit the HMTAb91-B lineage and was also the only contig from Sakinaw Lake that hit?

Figure S4: It states in the supplemental text that, Line 50 "The total percentage of Marinimicrobia represented in the 594 globally distributed metagenomes (fig. S4) ..." This figure looks to me to be contig length (y-axis) and only from 2 datasets (Northeastern subarctic Pacific (NESAP) Ocean and Saanich Inlet). Again in the main text it refers to fig S4 in: Line 128, "Metagenomic contigs 129 >5000bp and with >95% identity to SAGs were identified followed by tetra-nucleotide 130 frequency analysis to resolve specific clades (Fig. 3A, fig. S4)." And then continues talking about what is in fig S4 but then refers to Fig. 3A and B which is the "Energy metabolism of Marinimicrobia population genome bins". It appears that something is mixed up here either in the text or the figure/table references.

Very minor:

I noticed some instances where Fig had a period after (Fig.) and other instances where it was not present (Fig).

Reviewers' comments:

Reviewer #1 (Remarks to the Author):

Hawley et al. present an extensive omics analyses of an uncultivated bacterial lineage, Marinimicrobia. The analyses include single-cell genomics, metagenomics and metatranscriptomics and are primarily derived from oxygen minimum zones. The authors describe 10 distinct clades within the Marinimicrobia and try to identify metabolic interactions associated with each clade. Important findings in this study include the identification of a link between evolutionary diversification of Marinimicrobia and metabolic energy yields and their participation in C and N biogeochemical cycling. Finally, the authors propose that certain lineages of Marinimicrobia may express nitrous oxide reductase and thereby act as a sink for nitrous oxide in the oceans. In general, the analyses are well-done but I have strong reservations over the focus of the manuscript and especially some of the key findings.

Key concerns

My key concern is that the primary findings of a link between energy yields and the diversification of Marinimicrobia and the idea that they act as a global sink for N₂O is completely unsubstantiated. We know from a multitude of genomics studies that metabolic diversity is commonly observed in the environment, especially in microbial dark matter phyla. While I agree that this diversification could be linked to energy yields in specific niches, the authors provide no evidence of the same in Marinimicrobial lineages described in this manuscript. Part of the problem is associated with the treatment of omic approaches which is quite perplexing and could be improved. As an example, N₂O is identified as a key electron acceptor in the OMZs in two lineages, SHBH1141 and ZA3312c-A. The best genome completion estimates for these two lineages are 57% and 78%, respectively. Consequently, it is quite likely that other parts of the denitrification pathway might have been missed due to these incomplete genomes. The use of single cell genome clusters for mapping metagenomic scaffolds in lieu of directly employing an unsupervised binning approach for metagenomes like Metabat is problematic in my opinion especially if the key question centers on determining the confidence of missed genes. While a link between a certain lineage and a particular redox pair like N₂O/N₂ likely exists as determined from metabolic gene predictions, it can only be validated with further proof of (a) association with N₂O (b) disassociation with other components of the reductive nitrogen cycle like NO₃, NO₂, NO. Finally, it is an entirely different argument to suggest that evolutionary diversification is driven by specific metabolic innovations that are associated with the geochemical gradients.

With advances in microbial ecology, there is a greater appreciation for the interconnectedness of microbial communities and metabolism. Overall, I think this study and the dataset itself are teeming with potential. Yet as things stand, the biogeochemical discoveries reported need further proof to be validated and accepted.

RESPONSE:

We would like to thank Reviewer 1 for pushing us to look more critically at our metabolic reconstruction in the Marinimicrobia. To address their concern that there is a lack of evidence to support the notion that diversification is linked to energy yields in specific niches we have further analyzed the Marinimicrobia population genome bins for evidence of other denitrification genes (further discussed below and in supplemental section ‘*Denitrification genes in Marinimicrobia*’) as well as reworded some claims to better reflect the findings.

Reviewer 1’s concerns that the incomplete nature of the SAGs within the *nosZ* carrying lineages ZA3312c and SHBH1141 may result in missing other genes in the denitrification pathway is addressed by more detailed description of methods as well as thorough discussion of genes in the denitrification pathway found in population genome bins for these two clades in the supplement (section ‘*Denitrification genes in Marinimicrobia*’). Firstly, claims of ZA3312c and SHBH1141 carrying only the nitrous oxide reductase gene without any other genes in the denitrification pathway is based on analysis of the population genome bins rather than the incomplete SAGs. Estimated percent completion of the population genome bins are 95.8% and 91.7% for ZA3312c and SHBH1141 respectively, and the respective size of the population genome bins are 11 Mbp and 65.6 Mbp (Table S4). Given the estimated completion and size of the population genome bins it would be unlikely to miss additional genes in the denitrification pathway. Further analysis using both IMG/M pipeline annotations, MetaPathways annotations and BLASTp comparisons to RefSeq (February 2017) are further described in the supplement to support the lack of evidence of any additional denitrification genes within these two lineages. Additional discussion about many microbial lineages that carry only the *nosZ* gene and accessory proteins is provided in the supplement as well as data on any possible denitrification genes in all of the SAGs and population genomes.

In response to Reviewer 1’s concerns about our binning approaches we have elaborated on our methodology and use of CheckM outputs in the SAG-supervised binning approach and validation of generated population genome bins. Specifically, we have clarified and elaborated on our omics approaches for construction and validation of Marinimicrobia population genome bins that we believe strike an important balance between sensitivity and precision. As per Reviewer 1’s recommendation we elaborate on our supervised approach to binning with particular attention paid to the use of linear discriminant analysis (LDA) in the construction of population genome bins. LDA is an example of a supervised classifier, a form of machine learning algorithm (Libbrecht and Noble. ‘Machine learning applications in genetics and genomics’. Nature Reviews Genetics, 16: 321-332, 2015). Classifiers are used to attribute labels to data points. In this work, LDA is used to classify metagenomic sequences to bins. Supervised classification algorithms use one data set to train classification protocols for another data set. In this work, LDA is trained using the single-cell amplified genome (SAG)(sequence data and kmer frequency analysis), entirely informing the classification protocol of metagenomic contigs. Effectively, LDA is given an example, a SAG(s), of how to classify, thereby informing the classification protocol, and enabled classification of metagenomic sequences into bins.

In addition to this explanation we have carried out a cursory binning experiment using METABAT with three metagenomes from the SI dataset that were collected concurrently with samples used to generate the Saanich Inlet SAGs (BioSamples SAMN05224439, SAMN05224444, SAMN05224474). Due to the different methodologies between SAG-supervised binning and METABAT a direct comparison is not possible. METABAT uses coverage information (taking a .fastq as input), thus only a single metagenome can be processed alone, inclusion of reads (or sequences) from a SAG would not yield useful results as the coverage information in the SAG sequence data would differ tremendously from the metagenome based on multiple-displacement amplification and coverage bias. Thus, from the genome-bins produced by METABAT (default settings) we used sequence homology searches to the SAG sequences (via BLASTn 95%ID over 5000 bases, as was used in SAG-supervised analysis) to enable some comparison to the SAG-supervised binning approach used to create the Marinimicrobia population genome bins. Results indicated that of the bins that had hits to Marinimicrobia (13 out of 40 total bins for three metagenomes) three consisted of metagenomic contigs that hit to SAGs of multiple geographic origins (e.g. Saanich Inlet 100m and Gulf of Maine 1m) while the remaining 10 consisted of metagenomic contigs that hit to SAGs collected from multiple depths in Saanich Inlet as well as LineP. Comparison of the Marinimicrobia lineages (or SAG-clusters) which the METABAT bin metagenomic contigs hit to showed three of the METABAT bins consisted of contigs that only hit to a single SAG-cluster while the remaining 10 consisted of contigs that hit to multiple SAG-clusters (e.g. ZA3312c-A and Arctic96B-7-A or ZA3312c-A, SHAN400 and SHBH1141).

The checkM statistics of the METABAT bins yielded seven bins with >95% percent completion but strain heterogeneity scores (essentially an average percent similarity of redundant single copy marker genes within the bin) were very low for these bins (between 5 and 50) (see table 'Response to reviewers Table 1' below), indicating that these bins are likely constructed of multiple populations of more distantly related lineages. When compared to the CheckM statistics for our SAG-supervised binning, strain heterogeneity scores are >70% for 5 out of 7 population genome bins (with 100 representing 100% amino acid identity of redundant single copy marker genes within the bin) indicating that the genetic content of these bin originated from more genetically cohesive populations than reflected in the METABAT analysis. Taken together, while the METABAT results successfully assembled population genome bins with high completion scores they also manifested strain heterogeneity scores indicative of increased genomic divergence. This increases the potential for false discovery within bins that sample across multiple lineages. With our SAG-supervised binning approach we can bin (or classify) multiple metagenomes all together to create robust population genome bins with direct links to specific SAG(s), providing us with the sequence data upon which to assess metabolic potential and energy metabolism of a given clade at the population level.

Reviewer 1's suggestion that the link between ZA3312c and SHBH1141 lineages and utilization of N₂O/N₂ redox pair hinges on association with N₂O is not fully accurate, as NosZ consumes N₂O, which is typically present in nmol concentrations in OMZs [Torres-Beltran 2017 and Trimmer 2016]. NosZ activity would correlate with a decrease in N₂O concentration as observed in figure 4A where *nosZ* transcripts from SHBH1141 correlate with an N₂O trough at 120m July 2010, and 135m August 2010. Additional

discussion based on evidence of any denitrification pathway genes in the SAGs and population genome bins have been added to the supplement and table S5, showing that ZA3312c and SHBH1141 carry only the *nosZ* gene (and accessory proteins) and no other denitrification pathway genes.

Other concerns:

Lack of context in the introduction: Much of the introduction is spent talking about thermodynamics and the evolution of the term “eco-thermodynamics”. I would have appreciated more background and context especially in line with previous findings of the abundance and sulfur cycling in Marinimicrobia.

RESPONSE:

There is very little published information on the Marinimicrobia phylum as there are no cultured representatives and only the single-cell amplified genomes presented here and previously in Rinke et al. 2013 and Nobu et al. 2014, as well as a few environmental DNA fosmids (Wright et al. 2014). We have elaborated as much as possible on known metabolisms from these two studies in the introduction.

Genome reduction and streamlining: This is an interesting idea and is recurring mention throughout the manuscript. Fig. S1 comparing genome reduction in Marinimicrobia with other organisms is smart. However estimating these numbers from incomplete SAGs raises doubts about their accuracy. This could be benchmarked by plotting results from a number of SAGs with varying genome completions and checking for the slope of the line.

RESPONSE:

We have better articulated our methods for generation of figure S1 in the supplement and have included below additional figures of COG-functions, COG-based genome redundancy and percentage coding bases all verses genome (ie SAG) completeness for each SAG (Response to reviewer Figure 1). The main message from this figure is that we estimate consistent gene redundancy and percent coding bases for each MGA clade regardless of genome completeness. Therefore, we can validly conclude that clades like ZA3312c have low gene redundancy and high percent coding bases even though we only have partial genomes to refer to. Meanwhile, as a reference, we provide a graph for observed COG functions to show that increasing genome completeness does influence the total number of observed COG functions.

Genome completion: My understanding of checkM is that it uses lineage-specific marker genes. Considering that very few Marinimicrobia genomes existed in public databases prior to this study, an alternative set of single-copy genes to determine genome completion would be useful.

RESPONSE:

CheckM determines the lineage specific set of single copy marker genes to use based on the placement of detected single copy marker genes in phylogenetic trees. This can be

problematic when trying to assign dark matter groups to trees that are not represented in public databases. CheckM has surveyed 3,324 high-quality draft genomes to build a robust inventory of lineage specific marker gene inventories. Furthermore, CheckM reports the lineage used for a given genome for user information and evaluation of appropriateness. We have added the lineage-specific marker information for the population genome bins (Table S4) for the reader's information. In both cases the lineage-specific markers were at the level of root or bacteria, markers that are common to all known life or to all known bacteria.

HyaAB – Fig S8.b shows the hydrogenase operon structure. This seems to be a rather unusual operon with *psrC* and *psrB* in the midst of *hyaAB*. The authors should check for the other components of the hydrogenase system. Additionally, a tree of *hyaAB* genes might also be useful.

RESPONSE:

Thank you for noticing this, indeed further evaluation of this operon has revealed it to be dedicated to hydrogen metabolism and the *psrB* and *psrC* to actually be HybA NiFe hydrogenase small subunit and HybB NiFe hydrogenase cytochrome b respectively. Figure S8b has been corrected accordingly as well as discussions about ramifications of reversibility of hydrogen metabolism carried out by HybOABCD rather than Hya.

Cometabolism – Co-metabolism is a rather unusual word to use to describe the presence of multiple metabolisms. Cometabolism is more commonly used to determine the auxiliary degradation or transformation of a metabolite by a particular enzyme.

RESPONSE:

While we recognize the existing use of co-metabolism the authors feel strongly that co-metabolism is the best term to describe the proposed interactions of these microbial groups. The sulfur and nitrogen based molecules are in all likelihood originally from organic molecules and thus the proposed metabolic interactions are essentially an extension of degradation of other metabolites albeit in the context of nitrogen-based energy metabolism.

Use transcriptomics to link organisms – The authors primarily link organisms to each other using data derived from metabolic genes. While this is a good start, the presence of transcriptomic data for other organisms (Planctomycetes, Nitrospina, SUP05) as shown in Fig. S6 might be more useful to nail down some of the identified metabolic networks.

RESPONSE:

Expression of pertinent genes from these groups is found in figure S6b.

Line 99 – 101: I think this is highly speculative. To suggest that branching patterns actually coincide with energy yields, the authors would need to show that the redox pair identified with the particular group in Fig.1 is in fact the primary mode of metabolism in the environment for that organism.

RESPONSE:

We have modified some of our claims to reframe as hypotheses and also shown a more detailed inventory of genes in the denitrification pathway for all groups, indicating the determined use of electron acceptors is extensive and correct.

Focus on sulfur metabolism – Considering the authors have reported psrABC in Marinimicrobia in a previous study and no other sulfur genes were identified, the findings of sulfur cycling are not novel and could be toned down a little in the abstract and conclusions.

RESPONSE:

We agree that while the Marinimicrobia psrABC is not a novel discovery as it was first discussed in Wright et al. 2014, the findings in the current manuscript focus largely on the metabolic capacity, including polysulfide reductase, of different Marinimicrobia clades identified in this work as well as the global distribution of these clades and thus these metabolisms, topics not addressed by Wright et al. Given the constraint of the polysulfide reductase metabolism to Arctic96B-7-A/B, SHAN400 and SHBH1141 clades and global distribution of these clades generated in this work the statement of Marinimicrobia filling novel metabolic niches in the nitrogen and sulfur cycles is legitimate. This is particularly true with respect to metabolic between these cycles as described in the dynamic mutualism model.

Title – The title as it stands is extremely vague and misleading. The authors should consider modifying it to make it more meaningful.

RESPONSE:

Thank you for your input to our title, the authors feel it properly engages the reader in the study and is appropriate for the manuscript contents. However, we welcome educated suggestions that capture the exciting observations put forward in the paper.

Line 236 – “application of economic principles to microbial ecology” I am uncertain as to what principles are referred to here.

RESPONSE:

Thank you, we have clarified this in the text.

Minor:

Line 77 – “obligate metabolic interactions” is a strong word. I would prefer “likely” or “dependence” in this scenario.

RESPONSE:

We have altered the text to read ‘appear to form obligate metabolic dependencies’ .

Fig.2 – Redox tower is somewhat misleading as to a reader it may seem that the tree correlates with the energy yields/redox pairs which is not the case. Please include statement saying redox pairs are colored by lineage in the caption.

RESPONSE:

We have removed Redox pairs from the initial caption and indicated the color coding is consistent figure 1.

Table S2 – “inventory” is spelt wrong.

RESPONSE:

Thank you, it has been corrected.

Line 141 – should read fig.4

RESPONSE:

Thank you, it has been corrected.

Figure legends and axes are upside-down in a number of figures (like in fig S9). Please check these again.

Reviewer #2 (Remarks to the Author):

The authors describe, in a multi-omic study, the phylogeny and metabolic framework within the Marinimicrobia phylum. The authors use these datatypes to draw links to co-metabolic interactions. While representatives within this phylum do not appear to be prevalent (<10% of all the metagenomes groups shown, Fig S3), their activities are interesting and possibly allow them to fill fundamental niches within the marine environment.

Major:

No where do I see a clear representation of the global metagenomes recruitment distinctly showing relative abundance globally. As this is a main point to the paper it would be good to have either a table or representation for clear reference. Table S3 does not seem to have good representation and Figure S3 also collapses the datasets by type, e.g., Coastal OMZ, Open Ocean etc. And, Figure 1C, while nice to show distribution does not adequately the relative abundance of these clades.

RESPONSE:

We have added figure S3B that shows the global relative abundance of the individual Marinimicrobia clades. Overall Marinimicrobia SHBH1141 present primarily in sulfidic waters seems to be the most abundant clade globally, followed by dysoxic clade Arctic96B-1-A and suboxic clade SHAN400.

Minor:

In the supplemental material methods section ‘Metagenome fragment recruitment for biogeography’ Line 65 it states that the contigs were assigned to the clade with greatest percent identity OR alignment length. What would happen if a contig had the the greatest percent identity to one clade and the alignment length to another. Where would it be

assigned. This seems ambiguous or I am missing something.

RESPONSE:

Yes, this was unclear, we have modified the statement to be more clear, contigs were assigned to clades based on percent identity then (in case of tie) to the clade with longer alignment length.

Table S3: This appears to be the numbers of contigs (Illumina) or reads (454) that hit each Marinimicrobia lineage from the different metagenome groups and also appears very low. Is this the correct number ... as in 1 sequence (which could be a single gene from Illumina – it's hard to tell the contig lengths here) hit the HMTAb91-B lineage and was also the only contig from Sakinaw Lake that hit?

RESPONSE:

That is correct, it is the number of fragments (ie assembled Illumina contigs or 454 reads), the contig lengths were not included in the table, though contig length is the data show in the global analysis figures. We have modified the table to reflect this. Yes a single fragment from the Sakinaw Lake metagenome was recruited to the HMTAb91-B lineage.

Figure S4: It states in the supplemental text that, Line 50 “The total percentage of Marinimicrobia represented in the 594 globally distributed metagenomes (fig. S4) ...” This figure looks to me to be contig length (y-axis) and only from 2 datasets (Northeastern subarctic Pacific (NESAP) Ocean and Saanich Inlet). Again in the main text it refers to fig S4 in: Line 128, “Metagenomic contigs 129 >5000bp and with >95% identity to SAGs were identified followed by tetra-nucleotide 130 frequency analysis to resolve specific clades (Fig. 3A, fig. S4).” And then continues talking about what is in fig S4 but then refers to Fig. 3A and B which is the “Energy metabolism of Marinimicrobia population genome bins”. It appears that something is mixed up here either in the text or the figure/table references.

RESPONSE:

Thank you for noticing these discrepancies, we have modified the text accordingly.

Very minor:

I noticed some instances where Fig had a period after (Fig.) and other instances where it was not present (Fig).

RESPONSE:

Thank you for noticing these discrepancies, we have modified the text accordingly.

Response to reviewers table 1: CheckM analysis results of Metabat binning compared to SAG-supervised binning (IG-Supervised Binning)

Metagenomes used	Marinimicrobia population genome bin (lineage)	Marker Lineage	Estimated Completeness (%)	Strain heterogeneity	bin size (Mbp)	Number of single copy marker genes	Single Cell Genomes in Bin	Geographic origin of SAGs in bin
and NESAP	ZA3312c-A	root	95.8	94.33	11.0	56	AAA160-I06, AAA160-C11, AAA076-M08, AAA160-B08	Gulf of Maine 1m
and NESAP	ZA3312c-B	k_Bacteria	93.4	28	1.2	147	AAA0298-D23	HOT Station ALOHA 25m
and NESAP	HF770D10	k_Bacteria	41.2	100	1.4	104	AAA003-E22	South Atlantic Gyre 800m
and NESAP	Arctic96B7_A	root	100.0	70.67	50.9	56	AB-746_N13AB-902, AB-747_F21AB-903	Saanich Inlet 100m
and NESAP	Arctic96B7_B	k_Bacteria	96.6	63.36	6.0	104	AB-746_P06AB-902, JGI 0000113-D11	Saanich Inlet 100m, NESAP 2000m
and NESAP	SHAN400	root	87.5	99.64	32.2	56	AB-755_M21D07	Saanich Inlet 185m
and NESAP	SHBH1141	root	91.7	96.11	65.6	56	AB-750_L13AB-904, AB-755_E16C12, AB-751_D09AB-901	Saanich Inlet 150m and 185m
etabat Analysis								
Metagenome	Metabat bin	Marker Lineage	Estimated Completeness (%)	Strain heterogeneity	bin size (Mbp)	Number of single copy marker genes	SAGs with BLAST hits in Bin	Geographic origin SAGs with BLAST hits in bin
aanich Inlet August 2012, 100m	bin4096451.1	root	100.0	7.97	31.3	56	AAA160-B08, AB-746_N13AB_902, AB-746_P06AB_902, AB-747_F21AB-903	Gulf of Maine 1m, Saanich Inlet 100m
aanich Inlet August 2012, 100m	bin4096451.2	k_Bacteria	100.0	13.37	9.7	102	AAA160-B08, AB-746_N13AB_902, AB-747_F21AB-903	Gulf of Maine 1m, Saanich Inlet 100m
aanich Inlet August 2012, 100m	bin4096451.6	k_Bacteria	34.7	0	0.8	104	AB-746_N13AB_902, AB-747_F21AB-903	Saanich Inlet 100m
aanich Inlet August 2012, 150m	bin4096452.1	root	100.0	4.99	11.2	56	AAA160-B08, AB-750A02AB-903, AB-750L13AB-904, AB-751_D09BAB-904, AB-755_E16C12, AB-755_M21D07	Gulf of Maine 1m, Saanich Inlet 150m and 185m
aanich Inlet August 2012, 150m	bin4096452.2	k_Bacteria	96.6	11.65	3.3	104	AB-750A02AB-903, AB-755_M21D07	Saanich Inlet 150m and 185m
aanich Inlet August 2012, 150m	bin4096452.4	k_Bacteria	44.4	0	2.4	104	AB-750A02AB-903, AB-750L13AB-904, AB-751_D09BAB-904, AB-755_E16C12	Saanich Inlet 150m and 185m
aanich Inlet August 2012, 150m	bin4096452.7	k_Archaea	7.7	25	1.0	149	AB-750A02AB-903, AB-751_D09BAB-904, AB-755_M21D07	Saanich Inlet 150m and 185m
aanich Inlet August 2012, 200m	bin4096453.1	root	100.0	2.99	33.5	56	AB-746_N13AB_902, AB-746_P06AB_902, AB-747_F21AB-903, AB-750L13AB-904, AB-755_M21D07	Saanich Inlet 100m, 150m and 185m
aanich Inlet August 2012, 200m	bin4096453.2	k_Bacteria	98.3	28.6	14.2	104	AB-746_N13AB_902, AB-746_P06AB_902, AB-750A02AB-903, AB-750L13AB-904, AB-751_D09BAB-904, AB-755_E16C12, AB-755_M21D07	Saanich Inlet 100m, 150m and 185m
aanich Inlet August 2012, 200m	bin4096453.3	k_Bacteria	98.3	5.71	6.1	104	AB-746_N13AB_902, AB-755_M21D07	Saanich Inlet 100m and 185m
aanich Inlet August 2012, 200m	bin4096453.5	k_Bacteria	100.0	49.6	3.1	102	AB-755_M21D07, AB-750L13AB-904	Saanich Inlet 150m and 185m
aanich Inlet August 2012, 200m	bin4096453.9	root	0.0	0	0.7	56	AB-755_M21D07	Saanich Inlet 185m

Response to reviewer figure 1: Plots of COG functions, COG-based gene redundancy and percent coding bases vs percent genome completeness for Marinimicrobia SAGs.

Reviewers' comments:

Reviewer #1 (Remarks to the Author):

I have revisited the manuscript submitted by Hawley et al. and the authors to my opinion have responded to the most concerns that I raised in the previous review.

However, I do have one important unaddressed criticism with respect to the title. I understand that the authors feel strongly about the title but I also feel that this is extremely vague as it stands. As an example – the authors mention a number of the Marinimicrobia clades performing heterotrophy/respiration/amino acid fermentation (I assume this is partly stickland reactions) but these reactions are missing when it comes to their description of diversification of metabolism across clades, and in figures 1B and 2 thereby presenting a rather simplified scheme. I do not doubt the veracity of these findings but considering this manuscript focuses on a single phylum – I suggest the inclusion of “Marinimicrobia” in the title.

I prefer to not use LDA or any supervised algorithm for binning and would rather use an unsupervised coverage-based binner but at a certain level, this is a personal preference and I see no issues with the authors' arguments for using these methods.

Primarily, I am satisfied with their responses to major methodological questions involving binning and comparison of the population genomes, and potentially missing other denitrification genes. I am also glad that the authors have toned down some of their claims associated with energy metabolism in the manuscript.

To my opinion, it is publishable and I have no other major remarks for the authors.

Minor Concerns:

Fig.4 I suggest removing genes from this figure with no expression as this may be somewhat confusing to a reader (I understand the consistence in having the same 4 genes but I see no point in having them if they are absent)– e.g. narG is absent in SHBH1141.

Table S4: I suggest adding the contamination results from checkM for each of the bins to this table S4.

References are off: Supplementary Refs start at #31 instead of #32.

Reviewer #2 (Remarks to the Author):

The authors have successfully dealt with many of the reviewers comments/suggestions.

Reviewer 1 noted that Genome reduction and streamlining needed to be benchmarked. I agree. Figure S1 is nice however adding the additional benchmarking to supplemental and amending the manuscript text is needed. The figures provided in the response do show an increase in COG functions with increasing completeness (although there is a lot of spread in the data). Additionally, when looking at the %coding bases by genome completeness (which I'm guessing is not % but proportion) HMTAb91 has a SAG with nearly 80% genome completeness and the lowest coding % at ~91%. The spread in HMTAb91 for coding bases regardless of genome completeness is ~91-95. Is this significant? Also, the slope of the line would be negative with a decrease in coding bases ... the same is seen in SHBH1141 (with the exception of an outlier at genome completeness ~0.2. I don't feel this is enough data to conclude genome reduction and streamlining is occurring. More

concerning is that in the text, Line 147 it states that ZA3312c-A/B and HF770D10 are comparable to *Ca. Pelagibacter*, for ZA3312c-A/B this is correct (and in the response figures this clade does look like a candidate for genome streamlining, however in my copy it is not HF770D10 (yellow Fig S1) but ZA3648c (light orange, Fig S1) that resides near the *Pelagibacter* representations - the authors need to reevaluate this in the text. Further, Arctic96B-7 is stated as 'intermediate genome streamlining' however based on the plots in the response to reviewer the COG functions jumps (appearing significantly, and SHBH1141 does the same) with genome completeness - this could be a sign that incomplete SAGs are not useful for this clade or these results. I think the data does point to a case for ZA3312c, however it is fuzzy for the rest. Also, I'm not sure that merely stating for each clade 'intermediate genome streamlining' is helpful (or warranted) for the results.

Also based on the response to reviewer 1, I saw that when discussing the transcriptomics to link organisms ... Line 191-192 states that unlike Arctic96B-7, SHAN400 transcripts for ferredoxin, pyruvate:ferredoxin oxidoreductase, *psrAB* and *narG* were highly expressed in anoxic waters and refers to fig S9. There are a few things wrong with this statement and the figure. I do not see *psrAB* and *narG* listed in fig S9. Also, the text for the genes and x-axis label is upside down (still!). Finally, there is no indication that Arctic96B-7 expresses any ferredoxin/pyruvate:ferredoxin oxidoreductase at all, ever in this dataset as no bubbles are shown in fig. S9. Furthermore, Fig 4 referenced earlier shows that Arctic96B-7 and SHAN400 both, at all times, have similar expression levels of *narG* and *psrAB* and highest in anoxic waters (blue). Please clarify all.

From the rebuttal: No where do I see a clear representation of the global metagenomes recruitment distinctly showing relative abundance globally. As this is a main point to the paper it would be good to have either a table or representation for clear reference. Table S3 does not seem to have good representation and Figure S3 also collapses the datasets by type, e.g., Coastal OMZ, Open Ocean etc. And, Figure 1C, while nice to show distribution does not adequately the relative abundance of these clades.

RESPONSE: We have added figure S3B that shows the global relative abundance of the individual Marinimicrobia clades. Overall Marinimicrobia SHBH1141 present primarily in sulfidic waters seems to be the most abundant clade globally, followed by dysoxic clade Arctic96BA and suboxic clade SHAN400.

Fig. S3B does not show global distribution at all. This must be fixed. I'm guessing this is a plot of the lineage present in all samples - this does not give a global representation. Fig. 1C does the best job, however the list of total sequences and libraries searched is impressive - I wanted to see that breadth and depth of Marinomicrobia in all the libraries etc to get a sense of how much these organisms account for the different data points.

Line 208: 'positing a central role for SHBH1141 in OMZ N2O reduction globally'. Finding transcript activity in two datasets near each other does not give a global indication of N2O reduction.

Lines 193-194: This statement 'This is consistent with redox-driven niche 194 partitioning between Arctic96B-7 and SHAN400 clades in the Saanich Inlet water column.' Does not appear to be well supported. Please provide more data referenced etc. Also, they are within the same phylogenetic clade (sharing an ancestor) in Fig. 2. Please clarify.

Reviewer #1 (Remarks to the Author):

I have revisited the manuscript submitted by Hawley et al. and the authors to my opinion have responded to the most concerns that I raised in the previous review.

However, I do have one important unaddressed criticism with respect to the title. I understand that the authors feel strongly about the title but I also feel that this is extremely vague as it stands. As an example – the authors mention a number of the Marinimicrobia clades performing heterotrophy/respiration/amino acid fermentation (I assume this is partly stickland reactions) but these reactions are missing when it comes to their description of diversification of metabolism across clades, and in figures 1B and 2 thereby presenting a rather simplified scheme. I do not doubt the veracity of these findings but considering this manuscript focuses on a single phylum – I suggest the inclusion of “Marinimicrobia” in the title.

Thank you for taking a stand on this matter. We agree with your assessment and have modified the title to “Diverse Marinimicrobia bacteria mediate coupled biogeochemical cycles along eco-thermodynamic gradients”

I prefer to not use LDA or any supervised algorithm for binning and would rather use an unsupervised coverage-based binner but at a certain level, this is a personal preference and I see no issues with the authors’ arguments for using these methods.

Primarily, I am satisfied with their responses to major methodological questions involving binning and comparison of the population genomes, and potentially missing other denitrification genes. I am also glad that the authors have toned down some of their claims associated with energy metabolism in the manuscript.

To my opinion, it is publishable and I have no other major remarks for the authors.

Minor Concerns:

Fig.4 I suggest removing genes from this figure with no expression as this may be somewhat confusing to a reader (I understand the consistence in having the same 4 genes but I see no point in having them if they are absent)– e.g. narG is absent in SHBH1141.

Figure 4 has been modified to more clearly indicate which genes are expressed in a given clade.

Table S4: I suggest adding the contamination results from checkM for each of the bins to this table S4.

Thank you for raising this issue. CheckM calculates contamination based on the number of single copy marker genes, with the assumption that a given genome will have only a single copy or each gene. As the Marinimicrobia bins in this study were constructed from multiple independently assembled metagenomes there are multiple closely related copies of single copy marker genes within a given bin. This cofounds the contamination metric. Rather than reporting contamination we provide the strain heterogeneity metric produced by CheckM as a measure of

‘contamination’ or taxonomic divergence with a bin as described in the *Construction and validation of population genome bins* section.

References are off: Supplementary Refs start at #31 instead of #32.

We have corrected this error.

Reviewer #2 (Remarks to the Author):

The authors have successfully dealt with many of the reviewers comments/suggestions.

Reviewer 1 noted that Genome reduction and streamlining needed to be benchmarked. I agree. Figure S1 is nice however adding the additional benchmarking to supplemental and amending the manuscript text is needed. The figures provided in the response do show an increase in COG functions with increasing completeness (although there is a lot of spread in the data). Additionally, when looking at the %coding bases by genome completeness (which I’m guessing is not % but proportion) HMTAb91 has a SAG with nearly 80% genome completeness and the lowest coding % at ~91%. The spread in HMTAb91 for coding bases regardless of genome completeness is ~91-95. Is this significant? Also, the slope of the line would be negative with a decrease in coding bases ... the same is seen in SHBH1141 (with the exception of an outlier at genome completeness ~0.2. I don’t feel this is enough data to conclude genome reduction and streamlining is occurring. More concerning is that in the text, Line 147 it states that ZA3312c-A/B and HF770D10 are comparable to Ca. Pelagibacter, for ZA3312c-A/B this is correct (and in the response figures this clade does look like a candidate for genome streamlining, however in my copy it is not HF770D10 (yellow Fig S1) but ZA3648c (light orange, Fig S1) that resides near the Pelagibacter representations - the authors need to reevaluate this in the text. Further, Arctic96B-7 is stated as ‘intermediate genome streamlining’ however based on the plots in the response to reviewer the COG functions jumps (appearing significantly, and SHBH1141 does the same) with genome completeness – this could be a sign that incomplete SAGs are not useful for this clade or these results. I think the data does point to a case for ZA3312c, however it is fuzzy for the rest. Also, I’m not sure that merely stating for each clade ‘intermediate genome streamlining’ is helpful (or warranted) for the results.

Thank you for your attention to these details. We have added the genome streamlining benchmarking to figure S1 as fig S1B. There was previously an error in the estimated % completion calculation in ‘reviewer response figure 1’, the revised figure has been included in the supplement. This corrects the SAGs in clade HMTAb91 and SHBH1141 that previously showed high estimated % completion. There is an additional brief discussion in the supplemental information (lines 253-256) regarding benchmarking and remarks about ‘intermediate genome stream lining’ have been removed from the main text.

Thank you for noticing the discrepancy in the coloring and text of HF770D10 in figure S1, the text has been corrected.

Also based on the response to reviewer 1, I saw that when discussing the transcriptomics to link organisms ... Line 191-192 states that unlike Arctic96B-7, SHAN400 transcripts for ferredoxin,

pyruvate:ferredoxin oxidoreductase, psrAB and narG were highly expressed in anoxic waters and refers to fig S9. There are a few things wrong with this statement and the figure. I do not see psrAB and narG listed in fig S9. Also, the text for the genes and x-axis label is upside down (still!). Finally, there is no indication that Arctic96B-7 expresses any ferredoxin/pyruvate:ferredoxin oxidoreductase at all, ever in this dataset as no bubbles are shown in fig. S9. Furthermore, Fig 4 referenced earlier shows that Arctic96B-7 and SHAN400 both, at all times, have similar expression levels of narG and psrAB and highest in anoxic waters (blue). Please clarify all.

Thank you for noticing this error, the sentence was intended to address differences in the expression of Heme/copper-type cytochrome and NADH dehydrogenase from Arctic96B-7 and SHAN400. The text in line 201-204 has been changed accordingly.

Orientation of text on the x-axis in figure 9 has been fixed.

From the rebuttal: Nowhere do I see a clear representation of the global metagenomes recruitment distinctly showing relative abundance globally. As this is a main point to the paper it would be good to have either a table or representation for clear reference. Table S3 does not seem to have good representation and Figure S3 also collapses the datasets by type, e.g., Coastal OMZ, Open Ocean etc. And, Figure 1C, while nice to show distribution does not adequately the relative abundance of these clades.

We have added figure S3B that shows the global relative abundance of the individual Marinimicrobia clades. Overall Marinimicrobia SHBH1141 present primarily in sulfidic waters seems to be the most abundant clade globally, followed by dysoxic clade Arctic96BA and suboxic clade SHAN400.

Fig. S3B does not show global distribution at all. This must be fixed. I'm guessing this is a plot of the lineage present in all samples – this does not give a global representation. Fig. 1C does the best job, however the list of total sequences and libraries searched is impressive – I wanted to see that breadth and depth of Marinomicrobia in all the libraries etc to get a sense of how much these organisms account for the different data points.

We have included table S4 ‘Summary of relative abundance of Marinimicrobia lineages recruited from metagenomes’ that shows the percentage of the metagenome recruited to each Marinimicrobia lineage for each sample.

Line 208: ‘positing a central role for SHBH1141 in OMZ N₂O reduction globally’. Finding transcript activity in two datasets near each other does not give a global indication of N₂O reduction.

Text was revised to include detection of transcriptional activity in Saanich Inlet as well as the South Pacific (Peru and ETSP), warranting the claim that SHBH1141 may contribute to N₂O reduction in OMZs globally.

Lines 193-194: This statement ‘This is consistent with redox-driven niche

194 partitioning between Arctic96B-7 and SHAN400 clades in the Saanich Inlet water column.’ Does not appear to be well supported. Please provide more data referenced etc. Also, they are within the same phylogenetic clade (sharing an ancestor) in Fig. 2. Please clarify.

Thank you for noticing this inconsistency, the transcripts for the genes discussed here were incorrect and have been corrected to read ‘Arctic96B-7, SHAN400 transcripts for heme/copper-type cytochrome and NADH dehydrogenase were most highly expressed in anoxic waters (fig. S9A)’, which is consistent with the conclusion that there is redox-driven niche partitioning between these two clades in Saanich Inlet.

REVIEWERS' COMMENTS:

Reviewer #2 (Remarks to the Author):

The authors have examined and where necessary corrected the manuscript based on the comments and suggestions raised in the last review.